# Quantifying Intracellular Viral Pathogen: Specimen Preparation, Visualization and Quantification of Multiple Immunofluorescent Signals in Fixed Human Airway Epithelium Cultured at Air-Liquid Interface

**DOI:** 10.3390/jpm12101668

**Published:** 2022-10-07

**Authors:** Sharon L. Wong, Elvis Pandzic, Egi Kardia, Katelin M. Allan, Renee M. Whan, Shafagh A. Waters

**Affiliations:** 1School of Biomedical Sciences, Faculty of Medicine and Health, University of New South Wales, Sydney, NSW 2052, Australia; 2Molecular and Integrative Cystic Fibrosis Research Centre (miCF_RC), University of New South Wales, Sydney, NSW 2052, Australia; 3School of Clinical Medicine, Faculty of Medicine and Health, University of New South Wales, Sydney, NSW 2052, Australia; 4Katharina Gaus Light Microscopy Facility, Mark Wainwright Analytical Centre, University of New South Wales, Sydney, NSW 2052, Australia; 5Department of Respiratory Medicine, Sydney Children’s Hospital, Randwick, NSW 2031, Australia

**Keywords:** cystic fibrosis, airway infection, air-liquid interface, wholemount immunofluorescence, tiled imaging, quantification, SARS-CoV-2

## Abstract

Infection control and aggressive antibiotic therapy play an important role in the management of airway infections in individuals with cystic fibrosis (CF). The responses of airway epithelial cells to pathogens are likely to contribute to the pathobiology of CF lung disease. Primary airway epithelial cells obtained from individuals with CF, cultured and differentiated at air-liquid interface (ALI), effectively mimic the structure and function of the in vivo airway epithelium. With the recent respiratory viral pandemics, ALI cultures were extensively used to model respiratory infections in vitro to facilitate physiologically relevant respiratory research. Immunofluorescence staining and imaging were used as an effective tool to provide a fundamental understanding of host–pathogen interactions and for exploring the therapeutic potential of novel or repurposed drugs. Therefore, we described an optimized quantitative fluorescence microscopy assay for the wholemount staining and imaging of epithelial cell markers to identify distinct cell populations and pathogen-specific targets in ALI cultures of human airway epithelial cells grown on permeable support insert membranes. We present a detailed methodology using a graphical user interface (GUI) package to quantify the detected signals on a tiled whole membrane. Our method provided an imaging strategy of the entire membrane, overcoming the common issue of undersampling and enabling unbiased quantitative analysis.

## 1. Introduction

Cultures of human airway epithelial cells at air-liquid interface (ALI) conditions enable cellular differentiation and the formation of a polarized pseudostratified epithelium composed of major airway epithelial cell types (basal, secretory and ciliated cells). These ALI cultures closely resemble their in vivo morphology and function [1,2]. The inoculation of respiratory pathogens (bacteria, virus and/or fungus) in ALI cultures recapitulates key molecular mechanisms and dynamic interactions between the host and pathogens [3,4,5]. They are a valuable tool for disease modelling and drug discovery, particularly in cystic fibrosis (CF), whereby chronic lung infections caused by *Staphylococcus aureus*, *Pseudomonas aeruginosa*, respiratory syncytial virus (RSV), rhinovirus and *Aspergillus fumigatus* dominate and debilitate the lives of CF patients [6,7]. The utility of ALI cultures to model pathogen infections such as severe acute respiratory syndrome coronavirus 2 (SARS-CoV-2) have increased exponentially since the start of the coronavirus disease 2019 (COVID-19) in December 2019 [4,5,8,9,10,11].

Immunofluorescence (IF) staining is the method of choice in examining the cell morphology and subcellular localization of proteins [12]. IF relies on the binding of a primary antibody to the target protein. The primary antibody is either conjugated to a fluorophore (direct IF) or unconjugated (indirect IF), which then requires the subsequent binding of fluorophore-tagged secondary antibody to the primary antibody [13]. Depending on the biological question being addressed, the intensity and abundance of the fluorescence signal, the localization of the protein of interest and the IF-stained sample may be visualized by examination using various microscopy techniques.

Airway epithelial cells have demonstrated a reduced differentiation capacity with increasing passage [14]. IF staining is essential to visually confirm the defining characteristics of mature away epithelium by examining the expression and localization of epithelial cell markers in ALI cultures. In addition, IF staining enables the visualization of pathogen infection in situ, which could reveal morphological changes to epithelial cells post infection (remodeling) and the specificity of pathogens towards certain cell types (pathogen tropism). For instance, recent studies of SARS-CoV-2-infected ALI airway epithelium demonstrated the dynamics of SARS-CoV-2 infecting nasal and bronchial epithelial cells [4,15], where the destruction of the tight junction (ZO-1) of cells promoted the formation of multinucleated cells (syncytia lesions). In these in vitro studies, IF staining demonstrated the colocalization of SARS-CoV-2 viral proteins (nucleoprotein, spike) with ciliated cells and secretory (goblet and club) cells but not basal cells [4,16,17].

The wholemount staining of ALI cultures offers several advantages compared to transverse section staining (3–8 µm), whether they be paraffin-embedded or frozen in the optimal cutting temperature (OCT). For instance, in thin sections, proteins of interest expressed in low abundance, i.e., rare cell population ionocytes or a pathogen infection in a small number of cells, may be skipped, which could lead to misinformation. Wholemount staining is more time and cost effective, as staining can be limited to a single sample. Furthermore, imaging of the whole permeable support insert membrane (hereafter referred to as the membrane) of ALI cultures allows the unbiased quantification of the spatial distribution and intensity of the protein signal of interest. This also prevents the issue of undersampling commonly associated with quantification performed on a select few fields of view.

We describe a detailed stepwise protocol for the wholemount staining and imaging and the quantification of various epithelial cell markers and pathogens (such as the RNA virus SARS-CoV-2 antigen) in ALI cultures of CF and non-CF airway epithelial cells grown on membranes. The antibodies recommended in this protocol are well-established and were previously reported by our group [18,19] and by others (Appendix A). We extended this protocol with instructions for the quantification of the protein signal of interest. An analysis suite, which we developed for this purpose, is presented in a graphical user interface (GUI) package for ease of use. Our protocol can be used to visualize and quantify other pathogen infections in ALI cultures.

## 2. Materials and Methods

All resources used in this protocol are listed in Table 1.


**Isolation, expansion and differentiation of human airway epithelial cell cultures.**


Ethics approval to receive and culture human airway cells was approved by the Children’s Hospital Network Ethics Review Board (HREC/16/SCHN/120). All research was performed with written informed consent from participants or the guardian of participants.


**Airway epithelial cell conditional reprogramming expansion culture and differentiation at air-liquid interface.**


For the initial part of the protocol, including brushing the nasal inferior turbinate and expansion culture and freezing for biobanking, we referred to the previously published protocol described in detail in [20]. In short, the following steps were performed:(1)Human nasal epithelial (HNE) cells were collected into DMEM-based collection media with antibiotics following the brushing of the nasal inferior turbinate. Human bronchial epithelial (HBE) cells were collected from bronchoalveolar lavage fluid (BALF) obtained from participants during the bronchoscopy.(2)BALF or collection media containing nasal cells were spun at 300 g for 5 min at 4 °C.(3)Supernatant was removed, and the cell pellet was resuspended with 1 mL of conditional reprogramming (CR) media. Cell count was performed.(4)The airway epithelial cells were seeded into a Collagen-coated culture flask with 3T3-J2 irradiated feeder cells (seeding density of 8000 cells/cm^2^).(5)CR media was changed every second day until 90% confluency.(6)When cells reached 90% confluency, a differential trypsin method was used to dissociate the cells, and a cell count was performed.(7)Cells were cryopreserved for future use or were resuspended in PneumaCult Ex Plus Medium (antibiotic-free) for seeding directly onto membranes for differentiation.(8)Dissociated or thawed airway epithelial cells were plated onto the Collagen-coated apical compartment of permeable support insert(s) at a seeding density of 200,000–250,000 cells/insert in 150 μL Ex Plus Medium. Then, 750 μL Ex Plus Medium was added to the basal compartment below the inserts.(9)The changing of Ex Plus Medium was performed every second day until a confluent cell monolayer was formed (usually by day 4 post seeding).(10)Once confluent, cells were fed with PneumaCult ALI Medium (antibiotic-free) via both apical and basal compartments.

NOTE: Cells must be at least 95% confluence before switching from Ex Plus Medium to ALI Medium. If the void area persists after day 4 post seeding and the immediate surrounding cells look elongated (fibroblastic/growth arrest state), the area is unlikely to recover, and cultures can be aborted.

(11)After 2 days of culture in submerged ALI Medium, the air-liquid interface was created by removing the apical media and exposing the cells to air. ALI Medium was added to the basal compartment only.(12)The basal media was changed every second day until full differentiation (day 21–25 post ALI establishment).(13)Once per week, mucus that accumulated on the apical surface of the epithelium was removed by incubating 200 μL of warmed PBS in the apical compartment for 10 min at 37 °C and was subsequently removed by aspiration.

NOTE: Higher passage cells typically generate thinner cultures with lower transepithelial electrical resistance (TEER).


**Infection with pathogen.**


The method for the addition of the pathogen to cells (site of infection: apical or basal, multiplicity of infection, duration) varies depending on the characteristics and infectivity of the pathogen. Respiratory pathogens are commonly added to the apical compartment to mimic the exposure of the human airway to environmental insult. Representative images presented in this manuscript are of ALI cultures that received apical infection with SAR-CoV-2 for 72 h.

To preserve the epithelial barrier integrity, a maximum volume of 30 μL is recommended for chronic incubation in 6.5 mm membranes (24 well plate, 0.33 cm^2^). When inoculating with less than 30 μL, it is added to the center of the apical compartment with the pipette held at 90° to ensure no loss of inoculum. The inoculum should be checked to ensure it is not dispensed onto or left remaining on the wall of the permeable support insert after addition.

At the completion of the incubation time, the cultures should be visually inspected for leaks. If the epithelial barrier integrity of the cultures is compromised by pathogen infection, basal media will leak into the apical compartment. Before fixing ALI cultures for IF staining, other functional assays to assess the impact of pathogen infection can be performed:Measuring TEER by chopstick electrodes as described by [21];Imaging cilia beating frequency using high-speed light microscopy, as described by [20].


**Wholemount immunofluorescence staining.**


Day 1.


**Preparation.**


The correct selection and accurate preparation of buffers are essential to obtain consistent and reproducible results in sample preparation for immunofluorescence imaging. Table 2 outlines the buffers used in this protocol and their preparation and storage.

In this protocol, 4% PFA is the recommended fixative for all except for ZO-1, which is incompatible with PFA fixation. For ZO-1 staining, methanol:acetone is the recommended fixative [22].

NOTE:(A)Apply fixative, permeabilization and wash buffers to both apical and basal compartments to ensure that the buffers penetrate to all cells.(B)The volume of buffers, antibodies and mounting media in this protocol is optimized for 6.5 mm membranes (24 well plate, 0.33 cm^2^). A proportionally larger volume is needed for membranes with a larger surface area.


**Rinsing.**


(1)Remove basal culture media using a vacuum aspirator.(2)To remove cell debris, wash cells with room temperature (RT) PBS (200 µL apical, 1 mL basal) for 5 min at RT.(3)Remove PBS using vacuum aspirator.(4)Repeat PBS wash four times, for a total of five washes.


**Fixation.**


(A)4% PFA (all markers except ZO-1):
(1)Fix cells in RT 4% PFA (200 µL apical, 750 µL basal) for 30 min at RT.(2)Remove PFA using a vacuum aspirator.(3)Neutralize excess PFA with RT 100 mM glycine in PBS (200 µL apical, 750 µL basal) for 30 min at RT.(4)Remove neutralization buffer using a vacuum aspirator.(B)Methanol:Acetone (ZO-1):
(1)Fix and permeabilize cells in ice-cold methanol:acetone (200 µL apical, 750 µL basal) for 30 min at −20 °C.(2)Remove methanol:acetone using a vacuum aspirator.


**Permeabilization.**


(1)Permeabilize cells with RT 0.5% Triton-X in PBS (200 µL apical, 750 µL basal) for 30 min on ice.(2)Remove permeabilization solution using a vacuum aspirator.


**Blocking.**


(1)Wash cells with RT PBS (200 µL apical, 1 mL basal) to remove the residual fixative and permeabilization buffer.(2)Remove PBS using a vacuum aspirator.(3)Repeat PBS wash two times, for a total of three PBS washes.(4)Block cells with 10% normal goat serum in IF buffer (100 µL apical) for 90 min at RT. The basolateral compartment is kept empty.


**Primary staining.**


(1)Remove block buffer using a vacuum aspirator.(2)Incubate cells in the appropriate primary antibody(ies) (Table 1) diluted in block buffer (50 µL apical) for 48 h at 4 °C.

NOTE: Add 1 mL of purified water to the empty wells around the edge of the plate to prevent evaporation during the incubation period. The incubation period of 48 h is used to ensure antibodies penetrate to the basal cells. If the volume of the antibody required is less than 1 µL, take 1 µL of the antibody and perform dilution with IF buffer before adding to the sample. A pipetting volume of less than 1 µL introduces a larger margin of error.

(3)Seal the edges of the plate with sealing film (Parafilm) to prevent evaporation.(4)Store IF buffer in the fridge until secondary staining.

Day 2.


**Secondary and counterstaining.**


(1)Remove primary antibody(ies) using a vacuum aspirator.(2)Wash cells with RT IF buffer (200 µL apical) for 5 min at RT.(3)Remove IF buffer using a vacuum aspirator.(4)Repeat wash with IF buffer four times, for a total of five IF buffer washes.(5)Incubate cells in the appropriate secondary antibody(ies) (Table 1) diluted in block buffer (50 µL apical) for 3 h at RT. Add phalloidin and DAPI to the secondary antibody(ies) cocktail if staining for actin and nuclei, respectively.(6)Protect cells from light using aluminum foil.(7)After 3 h incubation, check fluorescent signal under a fluorescence microscope (such as EVOS fluorescence microscope with blue, green and red channels) to ensure cells have been appropriately stained.(8)Remove secondary antibody(ies) cocktail using a vacuum aspirator.(9)Wash cells with RT IF buffer (200 µL apical) for 5 min at RT.(10)Remove IF buffer using a vacuum aspirator.(11)Repeat wash with IF buffer four times, for a total of five IF buffer washes.


**Excising membrane.**


(1)Place the permeable support insert upright on a petri dish so that the membrane is touching the petri dish.(2)Use a fine scalpel (size 11) and cut around the membrane using the wall of insert as a guide (Appendix A).(3)Once the membrane is completely separated from the insert, lift the insert (without membrane) carefully and discard.

NOTE: Ensure the scalpel and/or insert does not come in contact with the cells to prevent the scratching of cells.

(4)Rinse membrane with IF buffer (200 µL) to remove any debris from the excision.(5)Wipe the IF buffer and any debris off the petri dish with low lint disposable wipes (Kimwipes). Repeat wash if needed.(6)Slide the scalpel (ensure no debris on scalpel) underneath the membrane and transfer the membrane, apical side upward-facing, to a microscope slide.

NOTE:(A)Ensure the membrane is free of debris before mounting on the microscope slide so there is no artefact and the membrane is flat for imaging.(B)The membrane can be cut into two or three equal sections to allow the staining of more protein markers. Cut the membrane after blocking and perform primary/secondary staining on a microscope slide with the cell side facing up.(C)If the membrane accidentally flips upside-down, the apical side can be identified by its matte appearance, while the basal side of the membrane is glossy.(D)If excising more than one membrane, do not let the membranes become dry. Residual buffer from a wet membrane helps the membrane to “cling” to the slide. A dry membrane has the tendency to curl up at the sides, increasing the chance of flipping upside-down.(E)Up to two membranes can be mounted on a microscope slide.


**Mounting.**


(1)Pipette 40 µL Vectashield Plus Antifade Mounting Medium onto the membrane without touching it.(2)Place a coverslip #1 thickness (0.13–0.16 mm) over the membrane at a 45° angle (one end resting against the slide and the other end supported with the scalpel) and lower the coverslip slowly to prevent air bubble formation.

NOTE: Lowering the cover slip at an angle instead of dropping it flat on the membrane pushes air outwards and reduces the likelihood of trapping air bubbles.

(3)Press the coverslip gently to remove excess mounting media and wipe clean with a low lint disposable wipe (Kimwipe).(4)Seal the edges of the coverslip with a coverslip sealant such as nail polish and let it dry.(5)Label the slide.(6)To ensure that the membrane is pressed flat optimal for imaging, sandwich the mounted membrane by first placing a disposable wipe (Kimwipe) and then a second slide (without sample) on top of the membrane slide. Align the slides so that when pressed, the weight is evenly distributed.(7)Cover the slides with aluminum foil to protect from light.(8)Press the slides with a weighted object, e.g., a full microscope slide box, and leave overnight at RT.

NOTE: Ensure the disposable wipe and foil are not folded or wrinkled so the same pressure is applied to the whole membrane.

(9)Store slides at 4 °C (protected from light) until imaging. Slides can be imaged immediately after being pressed or can be stored for at least one month at 4 °C.

Day 3.


**Imaging—confocal and epifluorescence.**


First, image the mounted membrane on a confocal microscope (such as Leica SP8 DLS confocal microscope) to visualize and validate stained protein targets. Second, image the same membrane on an epifluorescence microscope (such as Zeiss Elyra PALM/SIM microscope) using the tiled setting to stitch 25 × 25 fields of view (6.4 mm × 6.4 mm) for the quantification of the stained protein markers on the whole membrane.


**Confocal imaging.**


(1)Set up the microscope: objective: 63×/1.4; bit depth: 12-bit.(2)Set up the excitation laser and detector to optimize the collection of the fluorophores in the sample.(3)Adjust the format and/or zoom factor to fulfill the Nyquist sampling requirement [23]. If imaging a Z-stack, adjust the Z step size to fulfill the Nyquist sampling requirement.


**Expected outcomes.**


Airway epithelial cells differentiated on membranes at ALI consisted of the major epithelial cell types found in the human lung epithelium, including basal cells (p63, red), ciliated cells (acetylated tubulin, green) and goblet cells (MUC5AC, yellow) (Figure 1). The cultures also demonstrated a preserved epithelial barrier integrity with robust tight junction ZO-1 (grey, Figure 1).

ALI cultures inoculated with pathogens such as coronaviruses (CoVs) can be visualized by staining viral replicative intermediate, double-stranded RNA (dsRNA, stain all RNA viruses, Figure 2A) or viral proteins (nucleoprotein, spike, membrane) (Figure 2B). The co-staining of epithelial cell markers allowed the visualization of viral tropism towards specific epithelial cell types, e.g., ciliated and goblet cells (Figure 3A,B).


**Epifluorescent tiled imaging of whole membrane.**


(1)Position the center of the membrane in the center of the field of view.(2)Set up the microscope: objective: 20×/0.5; format: 512 × 512 pixels; bit depth: 16-bit.(3)Set up the 25 × 25 tile scan function (6.4 mm × 6.4 mm) to image the whole membrane.(4)To ensure the whole membrane aligns and fits within the 25 × 25 tile scan, acquire a preview image of the center five columns (5 × 25 tile scan setting) and rows (25 × 5 tile scan setting). Adjust the position of the membrane as needed.(5)Use the plane of focus at the center of the membrane as the plane of focus for the whole membrane.(6)Adjust the exposure time and detector gain of each channel based on the brightest sample, ensuring that saturation does not occur in any pixel throughout the sample.(7)Acquire one 25 × 25 tile scan image per channel.

NOTE: Stitched tile scan imaging of the whole membrane provides a “bird’s-eye” view of the pattern and coverage of viral infection, which facilitates the quantification of the signal in an unbiased manner.


**Quantification of whole membrane using the Graphical User Interface ‘ALIIFAnalysis’.**


To characterize the morphological and intensity properties of SARS nucleoprotein positive clusters, an analysis suite was developed in MATLAB (MathWorks, Natick, MA, USA). The analysis consisted of background illumination correction, followed by the intensity thresholding to define the SARS positive clusters, size exclusion on resulting binary image to filter out small and spurious clusters and finally exporting the cluster properties to an excel file. A Graphical User Interface (GUI) ‘ALIIFAnalysis’ was developed to facilitate this analysis. The GUI files and a user guide on how to package and install the GUI files together with the supporting scripts and the Open Microscope Environment (OME) package for MATLAB are available via GitHub at https://github.com/ElvPan/ALIIFAnalysis (accessed on 29 July 2022).

To analyze data, open the ALIIFAnalysis GUI interface in MATLAB (Figure 4A).

NOTE: Do not change the size or maximize the GUI interface, as this alters the display size of the loaded image.

**Figure 4 jpm-12-01668-f004:**
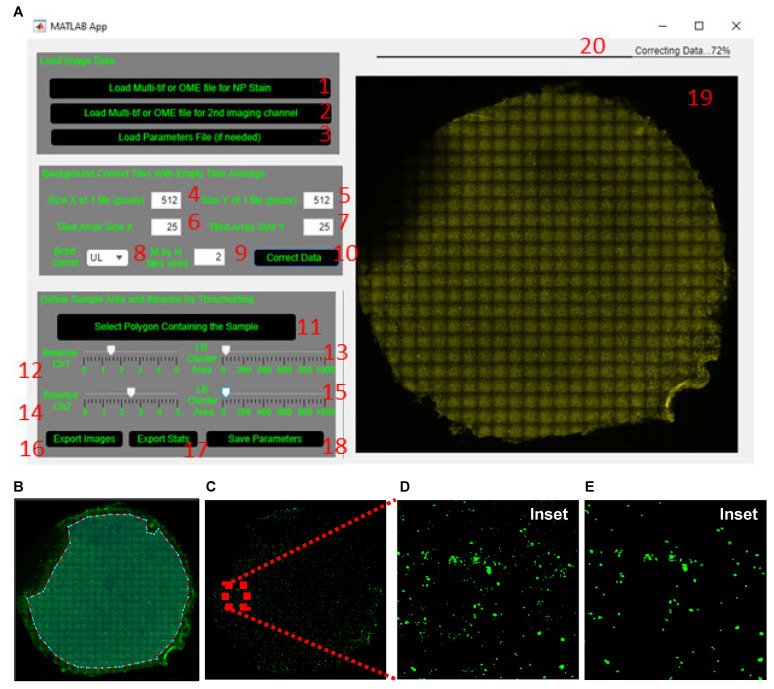
Quantification of SARS nucleocapsid protein signal in air-liquid interface cultures. (**A**) Graphical User Interface (GUI) ‘ALIIFAnalysis’ interface in MATLAB software. (**B**) Enclosed polygon outlining the region of the air-liquid interface (ALI) membrane for the analysis of the nucleocapsid protein (NP) signal. (**C**) Binary image of the NP signal. (**D**,**E**) Magnified inset of the NP signal (**D**) before and (**E**) after size exclusion.


**Loading image data.**


Sample raw image files and analyzed datasets are available via FigShare at https://figshare.com/articles/dataset/Sample_data_files_and_analysed_data_from_Sensitive_and_specific_immunofluorescence_protocol_for_pathogen_visualisation_and_quantification_in_fixed_human_airway_epithelium_cultured_at_air_liquid_interface/19882201 (accessed on 29 July 2022). 

(1)Load the tiled image for dsRNA or a pathogen-specific protein (SARS-CoV-2 nucleoprotein (NP) in example datafile) by clicking the ‘Load Multi-tif or OME file for NP stain’ button (labelled 1, Figure 4A), respectively.(2)Load the tiled image of a cell marker such as the acetylated tubulin or MUC5AC (or other imaged markers) by clicking the ‘Load Multi-tif or OME file for 2nd imaging channel’ button (labelled 2, Figure 4A). If no co-stain is added, skip this step.


**Background correcting tiles with empty tiles average.**


(1)Define the dimension of a single tile in pixels (512 × 512 pixels, labelled 4 and 5, Figure 4A) and the size of a tiled image (25 tiles × 25 tiles, labelled 6 and 7, Figure 4A).(2)Define which corner of the tiled image (upper left is default) and how many tiles within the chosen corner (2 by 2 is default) are used to obtain the average background tile profile (labelled 8 and 9, Figure 4A).(3)Apply the average background correction to every tile by clicking the ‘Correct Data’ button (labelled 10, Figure 4A).

NOTE: The average background profile from step 5 is normalized (divided) out from each tile in the composite image, resulting in an image with intensity values > 1. This step takes a while to process; the status of progress is shown above the loaded tiled image (labelled 20, Figure 4A). Do not proceed to the next step until data correction is complete. Once complete, the corrected tiled image is displayed (labelled 19, Figure 4A).


**Defining sample area and binarizing by thresholding intensity.**


(1)Manually exclude from the analysis the regions that are out of focus or auto-fluorescent or have artefact. To do this, click the ‘Select Polygon Containing The Sample’ button (labelled 11, Figure 4A). Once the crosshair cursor appears, select the region of the membrane to include for analysis. Once an enclosed polygon is obtained, double right click to end the polygon selection (Figure 4B).(2)Adjust the ‘Binarize Ch1’ slider (labelled 12, Figure 4A) to set the threshold for dsRNA or the NP signal to generate a binary image (Figure 4C,D).(3)Wait for the tiled image to update before re-adjusting the slider, since this step takes a few seconds to process (same for adjustment of the LB Cluster Area slider below).(4)Adjust the lower bound of the cluster area size using the ‘LB Cluster Area’ slider (labelled 13, Figure 4A) to filter out clusters smaller than ~10 µm (20 pixels in the sample image file provided). The inset in Figure 4C shows the same area of filter before (Figure 4D) and after (Figure 4E) cluster area filtering.(5)Proceed to the next step only after both ‘Binarize’ and ‘LB Cluster Area’ sliders have been adjusted; otherwise, the GUI will not generate the statistics excel in step 9.(6)Repeat steps 2–4 for acetylated tubulin or MUC5AC by adjusting the ‘Binarize Ch2’ and ‘LB Cluster Area’ sliders (labelled 14 and 15, Figure 4A).(7)Export the background corrected and final binary images for both channels (Figure 5A) by clicking the ‘Export Images’ button (labelled 16, Figure 4A).

NOTE: All files generated from the GUI ‘ALIIFAnalysis’ are exported to the same folder from which the raw image files were selected from.

(8)Proceed to the next step only after all images are successfully exported.(9)Export the statistics for each channel (Table 3, Figure 5B–D) by clicking the ‘Export Stats’ button (labelled 17, Figure 4A).(10)Wait for the export of statistics to finish before opening the excel file; otherwise, it will incur an error. Use the ‘Preview’ pane in File Explorer (Windows) or Finder (Mac) to check the progress of the data export without opening the file.

NOTE: A normalization step to the number of cells (nuclei staining) is not required for image analysis, as the data are expressed as percentage area coverage, not absolute area coverage (in pixel).

(11)Save the defined parameters by clicking the ‘Save Parameters’ button (labelled 18, Figure 4A) so that the same parameters can be applied to all tiled images from the same experiment.

NOTE: To apply the saved parameters to subsequent tiled images, load the images to analyze as outlined above and click the ‘Load Parameters File (if needed)’ button (labelled 3, Figure 4A).

(12)Representative stitched tiled images of acetylated tubulin, MUC5AC and NP signal are shown in Figure 6A,B.

**Figure 5 jpm-12-01668-f005:**
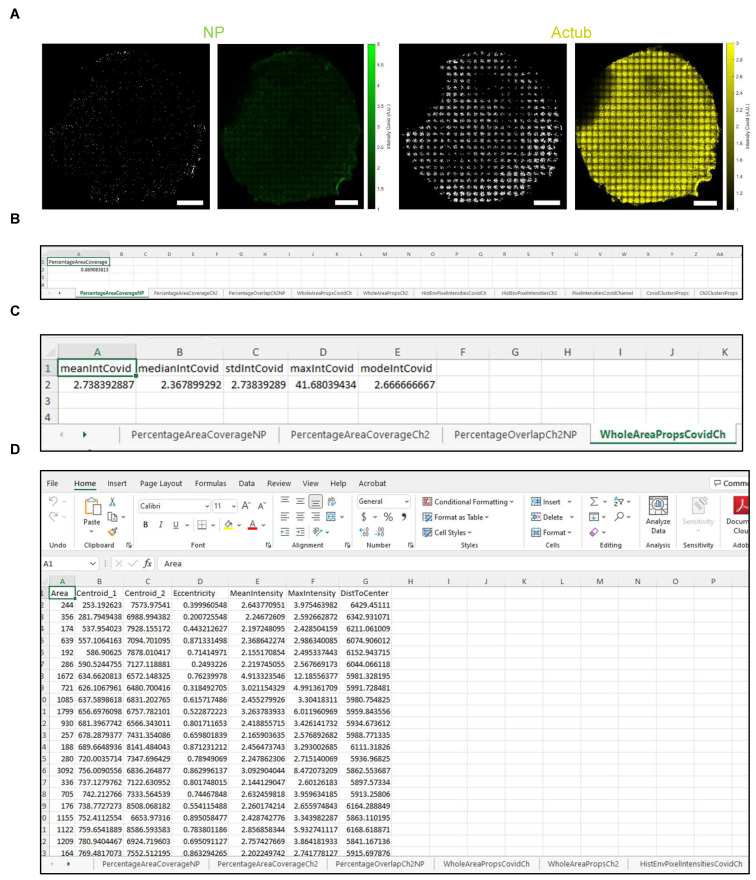
Representative graphical user interface analysis of data from CF air-liquid interface cultures. (**A**) Binary (left) and background corrected images (right) of SARS nucleocapsid protein (NP, green) and acetylated tubulin (Actub, yellow). Scale bars: 1 mm. (**B**) Area coverage (%) of NP signal. (**C**) Intensity statistics of the NP signal. (**D**) Statistics of the clusters after segmentation and filtering, for the NP and Actub signal.

**Table 3 jpm-12-01668-t003:** Example statistics generated from the GUI ‘ALIIFAnalysis’.

Tab Name	Output
PercentageAreaCoverageNP	Area coverage (%) of dsRNA or NP signal (channel 1).
PercentageAreaCoverageCh2	Area coverage (%) of acetylated tubulin or MUC5AC signal (channel 2).
PercentageOverlapCh2NP	Overlapping area (%) between dsRNA or NP (channel 1) and acetylated tubulin or MUC5AC (channel 2) signal.
WholeAreaPropsCovidCh	Intensity of dsRNA or NP signal (channel 1).
WholeAreaPropsCh2	Intensity of acetylated tubulin or MUC5AC signal (channel 2).
HistEnvPixelIntensitiesCovidCh	x and y envelopes of the histogram of intensity for NP-positive pixels
HistEnvPixelIntensitiesCh2	x and y envelopes of the histogram of intensity for acetylated tubulin- or MUC5AC-positive pixels
PixelIntensitiesCovidChannel	Intensity of all NP-positive pixels
CovidClustersProps	Statistics of the NP-positive clusters after segmentation and filtering. NOTE: Properties such as cluster area, centroid location and distance to filter center are expressed in pixels and need to be multiplied by pixel size to obtain physical units. Cluster eccentricity is unitless and is equal to 0 if the object is close to the circle and is 1 if it is line-like. The mean and max intensities of the cluster are expressed in relative fluorescence units.
Ch2ClustersProps	Statistics of the acetylated tubulin- or MUC5AC-positive clusters after segmentation and filtering

**Figure 6 jpm-12-01668-f006:**
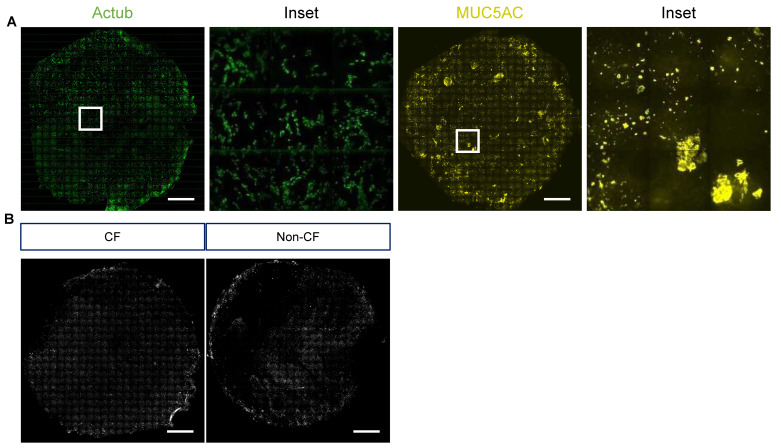
Representative stitched tiled images (25 × 25) of air-liquid interface cultures. (**A**) Acetylated tubulin (Actub) and MUC5AC with magnified insets (3 × 3). (**B**) SARS nucleocapsid protein (NP, grey) in SARS-CoV-2-infected air-liquid interface (ALI) cultures. A 20×/0.5 objective. Scale bars: 1 mm.

## 3. Discussion


**Fixative agent.**


Fixation preserves the cellular architecture and spatial organization of antigens, and prevents the degradation of the sample by inactivating proteolytic enzymes [24]. The aldehyde fixative, 4% PFA, is the most commonly used fixative for ALI cultures grown on membranes (Appendix A). This is not only because it best preserves cell morphology and structure but it is also compatible with most antigens. PFA is preferred over its monomeric form, neutral buffered formalin, as the latter usually has added methanol to prevent its polymerization [25], while PFA is methanol-free. Since methanol dehydrates and precipitates proteins, it can cause some loss of plasma membrane structure as well as organelle and nuclear content [26]. PFA works by crosslinking proteins; it reacts with primary amines in proteins to form partially-reversible methylene bridge crosslinks [27]. At times, this can change the molecular conformation of protein antigens and mask the binding site of antibodies, resulting in no or dim signal. This issue is more prevalent with monoclonal (single binding site) compared to polyclonal (multiple binding sites) primary antibodies. This can be resolved using either antigen retrieval (described next paragraph) or a coagulant fixative such as methanol and acetone, which fix cells by precipitating and denaturing proteins. For instance, this protocol uses a methanol:acetone fixative for the staining of tight junction ZO-1 due to a lack of consistent staining with PFA fixation, as reported previously [22]. Overfixation with formaldehyde can cause autofluorescence, as excessive reagent reacts with amines in proteins to generate fluorescent products [12]. No noticeable autofluorescence was observed using the volume and duration of PFA incubation recommended in this protocol. For troubleshooting autofluorescence, refer to Table 4.


**Antigen retrieval.**


Antigen retrieval (heat induced epitope retrieval or proteolytic induced epitope retrieval) is commonly applied to samples fixed with aldehyde fixatives to reverse the crosslinking of proteins, hence obtaining antibody labelling or improving signal intensity [28]. In this protocol, whole PFA-fixed membranes do not require antigen retrieval when stained with the listed antibodies. The use of antigen retrieval in wholemount staining of ALI cultures has not been reported in the literature [2,3,4,29]. Perhaps the antigens studied so far are resistant to crosslinking fixation by PFA or can otherwise be successfully labelled when fixed with coagulant fixative. If needed, antigen retrieval for ALI cultures grown on a membrane can be performed as described previously in thymic epithelial cells [30]. Should antigen retrieval be performed, optimization (pH, temperature, duration) is essential, and staining should be validated with a control sample not subjected to antigen retrieval to discriminate the artefact from the specific binding of the antibody. This is because antigen retrieval can increase both the specific and non-specific binding of antibodies and occasionally cause damage to the sample [31].


**Permeabilization.**


The permeabilization step is needed so that antibodies can travel through the plasma membrane and access the intracellular marker of interest. This step is not needed if the marker of interest is a cell surface marker or the sample has been fixed in coagulant fixatives with permeabilizing properties such as methanol and acetone. The detergent, Triton-X, is the most common permeabilizing agent for ALI cultures (Appendix A) and is a good starting point for optimization. Other detergents such as NP-40, saponin or Tween 20 may also be used.


**Blocking.**


Blocking buffer generally contains proteins such as normal serum (antibodies), bovine serum albumin (BSA), gelatin or skim dry milk, all of which have high affinity and hence bind to non-specific binding sites on samples [13]. These sites become inaccessible to antibodies, thereby reducing background staining. A low concentration of detergents such as Tween 20 is also added to blocking buffer to wash off non-specific binding. The blocking buffer used in this protocol is PBS with 10% normal goat serum, 0.1% BSA, 0.2% Triton-X and 0.05% Tween 20. The appropriate normal serum to use is one from the host species of secondary antibodies, not the primary antibodies. Otherwise, the secondary antibodies will detect the non-specific binding of the normal serum and generate a background signal. In this protocol, the primary and secondary antibodies are diluted in blocking buffer to enhance their specificity.


**Staining—primary and secondary.**


The quality of the antibody is critical to obtain specific and reproducible staining [32]. Primary antibodies used in this protocol were specifically chosen because their use was validated in ALI cultures in the literature (Appendix A). The antibodies include both polyclonal and monoclonal antibodies, but all have been affinity purified. In this protocol, only indirect antibody staining was used. The main advantage of indirect labelling is that the signal of the antigen can be amplified. It is also more economically viable, as it allows them to be used for other immunolabeling applications such western blotting, flow cytometry and enzyme-linked immunosorbent assay (ELISA). In addition, unconjugated antibodies offer more flexibility when choosing combination targets for multicolour IF staining.

All secondary antibodies used in this protocol were cross-adsorbed against immunoglobulins (IgG) from other species to reduce cross-reactivity and background staining. This protocol describes tri-colour staining (Figure 2 and Figure 3). More colours can be used depending on the availability of laser/filter on the microscope and fluorophores with no spectral overlap. As a different host species is needed for each primary antibody and each has its own secondary antibody, to accommodate for more multicolour staining, conjugated primary antibodies (direct staining) can be used to eliminate the need for secondary antibodies. When selecting secondary antibodies for multicolour staining, the Alexa Fluor 488 antibody is recommended for the least abundant protein, as it gives the brightest and most photostable signal. If an antibody other than the ones listed in this protocol is used, factors such as host species, clonality, purification, immunogenicity and species reactivity need to be considered and optimized. This is discussed in detail in [32]. It is important that a new antibody is validated with the use of appropriate controls (positive/negative, isotype, secondary antibody only).


**Mounting—mounting media and coverslip.**


This protocol recommends the use of a non-hardening mounting media without DAPI. The non-hardening solution is to facilitate the remounting of membrane, if necessary, as the mounting with a hardening mounting media is permanent. Mounting media with DAPI is not recommended, as the exposure of DAPI to ultraviolet excitation during imaging can contribute to autofluorescence as described in Table 4. A more detailed comparison of antifade reagents is available in [33].

This protocol recommends the use of standard #1 thickness coverslip (0.13–0.16 mm). The #1.5 thickness coverslip (0.16–0.19 mm) is often recommended, as most objectives are corrected to 0.17 mm thickness [34]. The #1.5 thickness is needed for samples that are directly in contact with the under-surface of the coverslip, i.e., cells cultured or blood cells smeared on the coverslip [35]. In this protocol, the membranes were overlaid with mounting media before being covered with a coverslip. The thickness of the overlaying mounting media has been reported to range between 5–30 µm, even following pressure being applied to remove excess mounting media [35]. The thickness of a #1.5 coverslip is similar to the thickness of a #1 coverslip plus the overlaid mounting media. Furthermore, ALI cultures are thicker than most adherent cells by at least 10 µm given the pseudostratified structure. Hence, this protocol recommends the standard #1 coverslip instead of the #1.5 coverslip.


**Imaging.**


If the signal of interest is a punctum (spot) or cannot be sufficiently resolved with the 20×/0.8 objective, higher numerical aperture objectives may be considered. If the signal of interest spans across different z-sections, the stitched tile scan image may need to be acquired at the different z-sections, with a confocal rather than epi-fluorescence imaging modality, to provide meaningful insight. Moreover, samples can be imaged on higher throughput microscopes such as slide scanners, which can handle a large volume of slides very quickly and reliably and image them in epifluorescence tiled mode.


**Analysis.**


The GUI ‘ALIIFAnalysis’ was implemented to characterize the properties of signal clusters in IF images of the whole membrane. It is based on principles of intensity thresholding and morphological operations on binary images to clean and extract the cluster properties. The GUI was provided to facilitate the analysis but can be used in other studies where IF data need to be quantified to assess the characteristic of clusters (organelles, cells) and extract their morphological properties (area, eccentricity, centroid, distance to some reference point such as a sample center) and their intensities (mean, sum, std) in the imaged channels. The GUI also allowed the calculation of the percentage of total sample area covered by NP-positive clusters. This analysis can be further adapted to quantify other available parameters such as the solidity, perimeter and convex hull area of clusters, should a particular study require comparison of these parameters. Moreover, while this protocol was applied to the 2D morphological analysis of clusters on 2D data sets, it can easily be extended to 3D data sets with equivalent functions readily available in MATLAB.


**Air-liquid interface cultures—in vitro cell model.**


ALI cultures are routinely used to study the epithelium of patients with chronic respiratory diseases. Nonetheless, it is important to acknowledge that some limitations exist since the in vitro culture environment is different to in vivo airway epithelium (extracellular matrix, lack of interaction with other cell types). These limitations are discussed in detail in previous studies [36,37,38,39].


**SARS-CoV-2 infection in CF and non-CF air-liquid interface cultures.**


To date, only one study has reported a comparison of SARS-CoV-2 infection in CF and non-CF primary air liquid interface cultures [40]. No change was observed in the SARS-CoV-2 viral load at 72 h post-infection in CF cultures, but, in non-CF cultures, a significant increase to >1 × 10^9^ copies/µL was observed. The transepithelial electrical resistance also remained unchanged in CF cultures but significantly decreased in non-CF cultures. No immunostaining was performed in this study, however.

**Table 4 jpm-12-01668-t004:** Troubleshooting.

Problem	Troubleshooting
Regions of the stitched tilescan image are out of focus.	This may be because the membrane is not flat or the cells appear as several layers due to the pseudostratified epithelial structure. Remounting the membrane may help if the membrane is not flat. Alternatively, acquire the stitched tilescan at multiple z-sections or manually outline and exclude the out of focus regions when performing quantification.
Fluorescent signal at the basal side (e.g., basal cell marker, p63) is dim and noisy.	This is likely because the basal side of the membrane is furthest away from the laser light source (most scattering and least penetration of light) and will have a lower signal-to-noise ratio. Use the Alexa Fluor 488 secondary antibody for a protein marker at the basal side to enhance the signal or perform tissue clearing to reduce scattering/increase the penetration of light. Ensure PBS wash is performed at least five times, five min each, to prevent non-specific binding.
Photobleaching	Photobleaching (fluorophore losing ability to fluoresce due to photochemical damage and covalent modification) is inevitable when imaging vertical Z-stack, considering the ALI cultures can be up to 50 µm thick. If a protein of interest has a dim signal and is localised to either the apical or basal half of the membrane, begin Z-stack imaging from that side of the membrane. Use detectors with higher sensitivity than the standard photomultiplier tubes (PMT) where available, such as the hybrid detector (HyD), which enables the use of reduced laser power and/or exposure time given its single-photon sensitivity and low dark noise (higher signal-to-noise ratio). Furthermore, ensure that photostable secondary antibodies and antifade mountants are used [33]. A comparison of antifade reagents was reviewed in detail in [33].
Autofluorescence	Dead cells are generally known to cause autofluorescence due to increased non-specific binding [41]. Extracellular matrix components such as collagen and elastin also often present with intrinsic fluorescence, contributing to autofluorescence in the sample [42]. Ensure that PBS wash is performed at least five times, five min each, to remove debris to prevent non-specific binding. Overfixing with aldehyde fixatives routinely generates autofluorescence, as excessive reagents react with amines and proteins to generate fluorescent products [12]. Prevent overfixing by using only enough volume and duration of 4% PFA to fix membranes. It is also possible to quench excess unreacted PFA with glycine [43]. Avoid using mounting media with DAPI, as the exposure of DAPI to just a few seconds of UV excitation light during fluorescence imaging can photoconvert the dye to stable green- and red-emitting forms [44]. Alternatively, solvents such as Eriochrome black T, Sudan black B and sodium borohydride have been shown to be effective in reducing autofluorescence in stained airway samples [42,45,46].

## Figures and Tables

**Figure 1 jpm-12-01668-f001:**
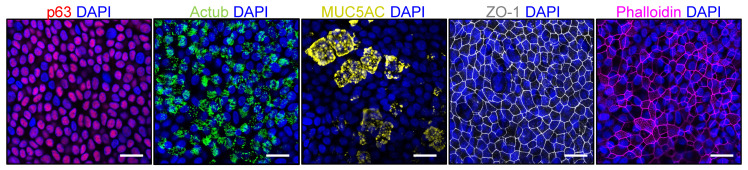
The characterization of airway epithelial cells differentiated at the air-liquid interface. Representative confocal images of acetylated tubulin (actub, ciliated cells), MUC5AC (secretory goblet cells), p63 (basal cells), ZO-1 (tight junction) and phalloidin (actin). A 63x/1.4 objective. Scale bars: 20 µm.

**Figure 2 jpm-12-01668-f002:**
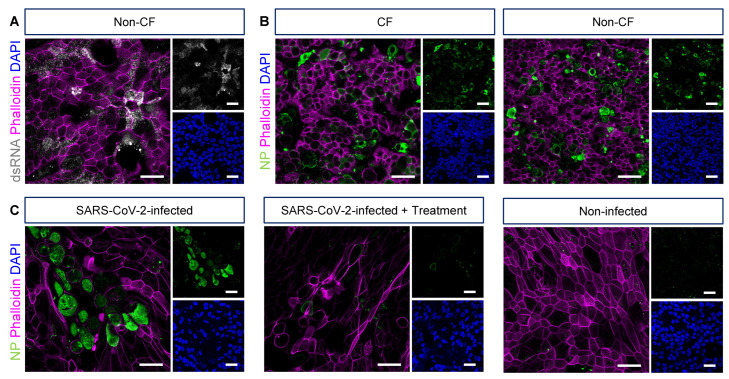
Visualization of pathogen infection in airway epithelial cell air-liquid interface cultures derived from CF and non-CF participants. (**A**) Representative confocal images of double-stranded RNA (dsRNA) in HCoV-OC43-infected air-liquid interface (ALI) cultures. (**B**) Representative confocal images of SARS nucleocapsid protein (NP) in SARS-CoV-2-infected ALI cultures. (**C**) NP signal in SARS-CoV-2-infected ALI cultures, with or without treatment with experimental compound, and in non-infected ALI cultures. A 63×/1.4 objective. Scale bars: 20 µm.

**Figure 3 jpm-12-01668-f003:**
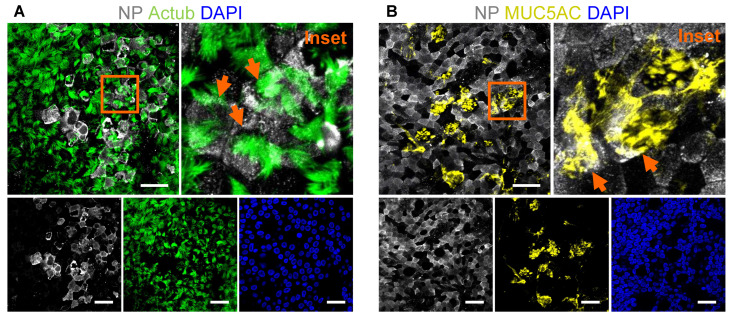
Visualization of pathogen tropism in air-liquid interface cultures. Representative confocal images of SARS nucleocapsid protein (NP) in SARS-CoV-2-infected air-liquid interface (ALI) cultures co-stained with (**A**) acetylated tubulin (actub, ciliated cells) and (**B**) MUC5AC (secretory goblet cells). Magnified insets show NP-positive ciliated and goblet cells (orange arrows). A 63×/1.4 objective. Scale bars: 20 µm.

**Table 1 jpm-12-01668-t001:** Key resources used in the protocol.

Reagent or Resource	Source	Identifier
**Antibodies|Dilution**		
Mouse monoclonal anti-dsRNA antibody J2|1:50	Jena Bioscience, Jena, Germany	RNT-SCI-10010200
Rabbit polyclonal anti-SARS Nucleocapsid Protein antibody|1:50	Novus Biologicals, Centennial, CO, USA	NB100-56576SS
Mouse monoclonal anti-Acetylated Tubulin|1:250	Sigma-Aldrich, St. Louis, MO, USA	T7451
Mouse monoclonal anti-MUC5AC (45M1)|1:250	Life Technologies, Carlsbad, CA, USA	MA5-12178
Rabbit monoclonal anti-p63 antibody [EPR5701]|1:100	Abcam, Cambridge, UK	ab124762
Rabbit polyclonal anti-ZO-1 antibody|1:100	Invitrogen, Waltham, MA, USA	61-7300
Goat polyclonal anti-mouse IgG antibody, Alexa Fluor 647|1:500	Life Technologies	A-21236
Goat polyclonal anti-rabbit IgG antibody, Alexa Fluor 488|1:500	Life Technologies	A-11034
Goat polyclonal anti-rabbit IgG antibody, Alexa Fluor 555|1:500	Life Technologies	A-21429
**Biological samples**		
Human nasal epithelial cells	Molecular and Integrative Cystic Fibrosis (miCF) Research Centre Biobank, Randwick, Australia	https://wch.med.unsw.edu.au/micf-research-centre (accessed on 29 July 2022)
Human bronchial epithelial cells	Molecular and Integrative Cystic Fibrosis (miCF) Research Centre Biobank	https://wch.med.unsw.edu.au/micf-research-centre (accessed on 29 July 2022)
3T3-J2 Irradiated feeder cells	STEMCELL Technologies, Vancouver, BC, Canada	100-0353
**Media/Supplements**		
Conditional Reprogramming (CR) Medium	STEMCELL Technologies	100-0352
PneumaCult™-Ex Plus Medium	STEMCELL Technologies	05040
PneumaCult™-ALI Medium	STEMCELL Technologies	05001
Hydrocortisone Stock Solution	STEMCELL Technologies	07925
Heparin Solution	STEMCELL Technologies	07980
PureCol^®^	Advanced Biomatrix, Carlsbad, CA, USA	5005
**Chemicals/Buffers**		
PBS, with Ca^2+^/Mg^2+^	Sigma-Aldrich	D8662
Phalloidin-Atto 565	Sigma-Aldrich	94072
4′,6-diamidino-2-phenylindole (DAPI)	Life Technologies	D1306
Vectashield Plus Antifade Mounting medium	Vector Laboratories, Burlingame, CA, USA	H-1900
16% Formaldehyde (*w*/*v*), Methanol-free	Thermo Fisher, Waltham, MA, USA	28908
Methanol	Ajax Finechem, Seven Hills, Australia	AJA318
Acetone	Ajax Finechem	AJA6
Glycine	Sigma-Aldrich	G7126
Triton-X	Sigma-Aldrich	T8787
Tween 20	Sigma-Aldrich	P2287
BSA	Sigma-Aldrich	A2153
Normal goat serum	Sigma-Aldrich	G9023
**Other**		
Corning 6.5 mm Transwell with 0.4 μm pore polyester membrane insert	Sigma-Aldrich	CLS3470
Petri dish (60 mm)	Thermo Fisher	150326
SuperFrost^®^ Plus Slides	Thermo Fisher	MENSF41296SP
Glass cover slips #1, 22 mm × 50 mm	Thermo Fisher	MENCS22501GP
Scalpel size 11	-	-
Disposable wipes (Kimwipes)	Kimberly-Clark, Irving, TX, USA	-
**Software and algorithms**		
MATLAB v9.9.0.1467703	MathWorks, Natick, MA, USA	
Image J software v1.53c	National Institute of Health, Bethesda, MD, USA	
**Microscope**		
EVOS fluorescence microscope	Thermo Fisher	-
Zeiss Elyra PALM/SIM microscope	Carl Zeiss, Jena, Germany	
Leica SP8 DLS confocal microscope	Leica Microsystems, Wetzlar, Germany	

**Table 2 jpm-12-01668-t002:** Buffer preparation procedure.

Buffers	Storage
Fixative: 4% paraformaldehyde (PFA) 1 mL 16% Formaldehyde (*w*/*v*), methanol-free 3 mL PBS	Aliquot and store at −20 °C for up to 6 months. Avoid exposure to light.
Neutralisation buffer: 100 mM glycine in PBS 75.07 mg glycine 10 mL PBS	Prepare fresh.
Permeabilisation buffer: 0.5% Triton-X in PBS 50 µL Triton-X 10 mL PBS.	Store at 4 °C for up to 1 year.
Immunofluorescence (IF) buffer: PBS with 0.1% BSA, 0.2% Triton, 0.05% Tween 20 40 mg BSA 40 mL PBS 80 µL Triton-X 20 µL Tween 20	Store at 4 °C for up to 2 days.

## Data Availability

The sample data presented in this study are openly available in FigShare at https://figshare.com/articles/dataset/Sample_data_files_and_analysed_data_from_Sensitive_and_specific_immunofluorescence_protocol_for_pathogen_visualisation_and_quantification_in_fixed_human_airway_epithelium_cultured_at_air_liquid_interface/19882201 (accessed on 29 July 2022).

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
