# Peer review of "Quantifying Intracellular Viral Pathogen: Specimen Preparation, Visualization and Quantification of Multiple Immunofluorescent Signals in Fixed Human Airway Epithelium Cultured at Air-Liquid Interface"

_jpm, 2022, doi:10.3390/jpm12101668_

Round 1

Reviewer 1 Report

The authors provide a clear protocol for the analysis of whole membranes of fully differentiated and pathogen infected CF and non-CF airway epithelial cells cultured at ALI. I like the idea of using an entire membrane to get the ‘whole picture’ and not just a region of a filter.  Still, I have a few minor comments.

In the paragraph of blocking (step 4) with 10% normal goat serum in IF buffer (100 μl apical) for 90 min at RT, it is not clear if there is PBS in the basolateral side or if it is kept empty. This also applies to the following procedure described in the protocol.

Is there no normalization step to the number of cells needed when performing the image analysis (nuclei staining)? In particular, when excluding regions for analysis how is it normalized to the area or number of cells analyzed?

As the protocol uses primary cells from individuals with CF and non-CF it would have been nice to see some CFTR stainings and analysis using the presented interface.

Author Response

Reviewer 1 

The authors provide a clear protocol for the analysis of whole membranes of fully differentiated and pathogen infected CF and non-CF airway epithelial cells cultured at ALI. I like the idea of using an entire membrane to get the ‘whole picture’ and not just a region of a filter.  Still, I have a few minor comments. 

1.    In the paragraph of blocking (step 4) with 10% normal goat serum in IF buffer (100 μl apical) for 90 min at RT, it is not clear if there is PBS in the basolateral side or if it is kept empty.

Response: The basolateral compartment is kept empty during the blocking step, no PBS was added.

Action: We have amended Step 4 to clarify (line 206-207), the sentence now reads “Block cells with 10% normal goat serum in IF buffer (100 µl apical) for 90 min at RT. The basolateral compartment is kept empty.”

2.    This also applies to the following procedure described in the protocol. Is there no normalization step to the number of cells needed when performing the image analysis (nuclei staining)? In particular, when excluding regions for analysis how is it normalized to the area or number of cells analyzed?

Response: The number of cells for each insert was standardized when seeding (200,000-250,000 cells/insert). When performing image analysis, data was not normalized to nuclei staining. This also applies to when excluding regions for analysis. Our data is expressed as percentage area coverage, not absolute area coverage (in pixel). We believe that the rate of infection (proportion of cells infected) remains the same regardless of the area of insert analysed. Hence, normalization to the number of cells is not necessary even when excluding regions for analysis.

Action: We have added a note in the GUI ‘ALIIFAnalysis’ section to clarify (line 428-430). The note reads “NOTE: A normalization step to the number of cells (nuclei staining) is not required for image analysis as the data is expressed as percentage area coverage, not absolute area coverage (in pixel).”

3.    As the protocol uses primary cells from individuals with CF and non-CF it would have been nice to see some CFTR stainings and analysis using the presented interface.

Response: We agree visualizing and quantifying wholemount CFTR staining would be beneficial for research in CF. However, that was not the aim of this manuscript. There is no study to date which suggests SARS-CoV-2 infection modulates CFTR expression in ALI cultures. Furthermore, given most CFTR mutations radically reduce CFTR abundance on the cell surface, CFTR staining in previous studies showed low CFTR expression (dim signal) in CF cultures [1, 2]. Imaging and analysis of CFTR in whole insert may not be the most sensitive and reliable methods to quantify expression of CFTR protein.  

Action: No change to manuscript. 

Reference

1.    Awatade, N.T., et al., Significant functional differences in differentiated Conditionally Reprogrammed (CRC)- and Feeder-free Dual SMAD inhibited-expanded human nasal epithelial cells. J Cyst Fibros, 2021.
2.    Pranke, I.M., et al., Correction of CFTR function in nasal epithelial cells from cystic fibrosis patients predicts improvement of respiratory function by CFTR modulators. Sci Rep, 2017. 7(1): p. 7375.

3.    Zhu, N., et al., Morphogenesis and cytopathic effect of SARS-CoV-2 infection in human airway epithelial cells. Nature Communications, 2020. 11(1): p. 3910.
4.    Morrison, C.B., et al., SARS-CoV-2 infection of airway cells causes intense viral and cell shedding, two spreading mechanisms affected by IL-13. Proceedings of the National Academy of Sciences, 2022. 119(16): p. e2119680119.
5.    Hou, Y.J., et al., SARS-CoV-2 Reverse Genetics Reveals a Variable Infection Gradient in the Respiratory Tract. Cell, 2020. 182(2): p. 429-446.e14.
6.    Gamage, A.M., et al., Human Nasal Epithelial Cells Sustain Persistent SARS-CoV-2 Infection <i>In Vitro</i>, despite Eliciting a Prolonged Antiviral Response. mBio, 2022. 13(1): p. e03436-21.
7.    Lotti, V., et al., CFTR Modulation Reduces SARS-CoV-2 Infection in Human Bronchial Epithelial Cells. Cells, 2022. 11(8): p. 1347.

Reviewer 2 Report

The manuscript submitted to JPM by Wong et al descript methods of quantifying SARS-CoV infected to human airway epithelium cultured at the air-liquid interface (ALI). Human nasal epithelial cells (HNE) were isolated after brushing the nasal inferior turbinate, and human bronchial epithelial cells (HBE) were collected from bronchoalveolar lavage fluid during bronchoscopy. The HNE and HBE cells were initially cultured in conditional reprogramming (CR) media, and then transferred onto membrane inserts to differentiate in PneumaCult Ex Plus Medium. SARS-CoV-2 was infected apically followed by immunofluorescent microcopy using specific epithelial and viral markers. The methods were supposed to be used to recapitulate the biology of viral infection in human airway. The ALI culture might be a good in vitro model for infection study, perhaps the authors should discuss on the drawbacks of this tool comparing to the sectioning of the patient samples, how different epithelial cell population might have been changed after isolation, re-culture, and re-differentiation? A few minor questions: what exactly the microscopes had been used? How the viral infection affects the epithelium? What are the differences of CoV-2 infection comparing CF vs. non-CF epithelium?

Author Response

Reviewer 2 

The manuscript submitted to JPM by Wong et al descript methods of quantifying SARS-CoV infected to human airway epithelium cultured at the air-liquid interface (ALI). Human nasal epithelial cells (HNE) were isolated after brushing the nasal inferior turbinate, and human bronchial epithelial cells (HBE) were collected from bronchoalveolar lavage fluid during bronchoscopy. The HNE and HBE cells were initially cultured in conditional reprogramming (CR) media, and then transferred onto membrane inserts to differentiate in PneumaCult Ex Plus Medium. SARS-CoV-2 was infected apically followed by immunofluorescent microcopy using specific epithelial and viral markers. The methods were supposed to be used to recapitulate the biology of viral infection in human airway.  

1. The ALI culture might be a good in vitro model for infection study, perhaps the authors should discuss on the drawbacks of this tool comparing to the sectioning of the patient samples, how different epithelial cell population might have been changed after isolation, re-culture, and re-differentiation?

Response: We acknowledge some limitations exist for ALI cultures as an in vitro cell model.   

Action: We have added a paragraph about limitations of ALI cultures in discussion section (line 584-589). The paragraph reads “ALI cultures are routinely used to study the epithelium of patients with chronic respiratory diseases. Nonetheless, it is important to acknowledge some limitations exist since the in vitro culture environment is different to in vivo airway epithelium (extracellular matrix, lack of interaction with other cell types). These limitations are discussed in detail in previous studies [32-35].” 

2.    A few minor questions: what exactly the microscopes had been used?

Response: The microscopes used for both epifluorescence and confocal imaging are stated in the key resources table (Table 1). To ensure clarity, lines 231-232 and 293-296 have been altered to include the name of these microscopes in the body of the manuscript.

Action:

Lines 231-232 read:
7)    After 3 h incubation, check fluorescent signal under a fluorescence microscope (such as EVOS fluorescence microscope with blue, green and red channels) to ensure cells have been appropriately stained.

Lines 293-296 read:

First, image the mounted membrane on a confocal microscope (such as Leica SP8 DLS confocal microscope) to visualize and validate stained protein targets. Second, image the same membrane on an epifluorescence microscope (such as Zeiss Elyra PALM/SIM microscope) using the tiled setting to stitch 25 × 25 fields of view (6.4 mm × 6.4 mm) for quantification of the stained protein markers on the whole membrane.  

3.    How the viral infection affects the epithelium?

Response: In this protocol manuscript, we showed that SARS-CoV-2 could infect both ciliated cells and mucus-secreting goblet cells (Fig 3A-B). As this is a protocol manuscript, only a small number of samples were analysed hence no other conclusive observation can be made. The morphological and cytopathic effect of SARS-CoV-2 infection (syncytial cell formation, deformed cilia) were previously characterised in several studies [3-6].

Action: We have included a number of references in the introduction which the reader could refer to for better understanding of impact of viral infection on epithelium.
The sentence now reads: “The utility of ALI cultures to model pathogen infections such as severe acute respiratory syndrome coronavirus 2 (SARS-CoV-2) have increased exponentially since the start of coronavirus disease 2019 (COVID-19) in December 2019 [4, 5, 8-11].”

4.    What are the differences of CoV-2 infection comparing CF vs. non-CF epithelium?

Response: Considering this is a protocol manuscript, we imaged SARS nucleocapsid protein and acetylated tubulin in a small number of samples only, CF (n=1) and non-CF (n=2), hence a valid comparison cannot be made (Fig 6B). There is one publication so far which reported comparison of SARS-CoV-2 infection in CF and non-CF primary air liquid interface cultures [7].

Action: We have added details about the published study in the discussion section (line 591- 597). The paragraph reads: “To date, only one study has reported comparison of SARS-CoV-2 infection in CF and non-CF primary air liquid interface cultures [7]. No change was observed in the SARS-CoV-2 viral load at 72 hours post-infection in CF cultures but in non-CF cultures, a significant increase to >1 × 109 copies/µL was observed. The transepithelial electrical resistance also remained unchanged in CF cultures but significantly decreased in non-CF cultures. No immunostaining was performed in the study however.”

Reference

1.    Awatade, N.T., et al., Significant functional differences in differentiated Conditionally Reprogrammed (CRC)- and Feeder-free Dual SMAD inhibited-expanded human nasal epithelial cells. J Cyst Fibros, 2021.
2.    Pranke, I.M., et al., Correction of CFTR function in nasal epithelial cells from cystic fibrosis patients predicts improvement of respiratory function by CFTR modulators. Sci Rep, 2017. 7(1): p. 7375.

3.    Zhu, N., et al., Morphogenesis and cytopathic effect of SARS-CoV-2 infection in human airway epithelial cells. Nature Communications, 2020. 11(1): p. 3910.
4.    Morrison, C.B., et al., SARS-CoV-2 infection of airway cells causes intense viral and cell shedding, two spreading mechanisms affected by IL-13. Proceedings of the National Academy of Sciences, 2022. 119(16): p. e2119680119.
5.    Hou, Y.J., et al., SARS-CoV-2 Reverse Genetics Reveals a Variable Infection Gradient in the Respiratory Tract. Cell, 2020. 182(2): p. 429-446.e14.
6.    Gamage, A.M., et al., Human Nasal Epithelial Cells Sustain Persistent SARS-CoV-2 Infection <i>In Vitro</i>, despite Eliciting a Prolonged Antiviral Response. mBio, 2022. 13(1): p. e03436-21.
7.    Lotti, V., et al., CFTR Modulation Reduces SARS-CoV-2 Infection in Human Bronchial Epithelial Cells. Cells, 2022. 11(8): p. 1347.